# A Clinical Study on Urinary Clusterin and Cystatin B in Dogs with Spontaneous Acute Kidney Injury

**DOI:** 10.3390/vetsci11050200

**Published:** 2024-05-02

**Authors:** Emilia Gordin, Sanna Viitanen, Daniel Gordin, Donald Szlosek, Sarah Peterson, Thomas Spillmann, Mary Anna Labato

**Affiliations:** 1Internal Medicine Section, Department of Equine and Small Animal Medicine, Faculty of Veterinary Medicine, University of Helsinki, 00014 Helsinki, Finland; sanna.viitanen@helsinki.fi (S.V.); thomas.spillmann@helsinki.fi (T.S.); 2Department of Nephrology, Helsinki University Hospital, University of Helsinki, 00290 Helsinki, Finland; daniel.gordin@hus.fi; 3Minerva Institute for Medical Research, 00220 Helsinki, Finland; 4IDEXX Laboratories, Inc., One IDEXX Drive, Westbrook, ME 04092, USA; donald-szlosek@idexx.com (D.S.); sarah-peterson@idexx.com (S.P.); 5Department of Clinical Sciences, Cummings School of Veterinary Medicine, Foster Hospital for Small Animals, Tufts University, North Grafton, MA 01536, USA; mary.labato@tufts.edu

**Keywords:** acute kidney injury, biomarker, tubular, dog, urinary clusterin, urinary cystatin B

## Abstract

**Simple Summary:**

In our study, we looked into finding better ways to diagnose and predict acute kidney injury (AKI) in dogs. To do so, two substances were studied in dog urine, urinary clusterin (uClust) and cystatin B (uCysB), to see if they could be useful as markers for AKI. We compared samples from 18 dogs with different levels of AKI and 10 healthy dogs. Dogs with AKI initially had higher levels of uClust and uCysB than healthy dogs. Also, the level of uCysB was higher in dogs that did not survive. In conclusion, uClust and uCysB seem to be promising markers for diagnosing and predicting the outcome of AKI in dogs.

**Abstract:**

Novel biomarkers are needed in diagnosing reliably acute kidney injury (AKI) in dogs and in predicting morbidity and mortality after AKI. Our hypothesis was that two novel tubular biomarkers, urinary clusterin (uClust) and cystatin B (uCysB), are elevated in dogs with AKI of different etiologies. In a prospective, longitudinal observational study, we collected serum and urine samples from 18 dogs with AKI of different severity and of various etiology and from 10 healthy control dogs. Urinary clusterin and uCysB were compared at inclusion between dogs with AKI and healthy controls and remeasured one and three months later. Dogs with AKI had higher initial levels of uClust (median 3593 ng/mL; interquartile range [IQR]; 1489–10,483) and uCysB (554 ng/mL; 29–821) compared to healthy dogs (70 ng/mL; 70–70 and 15 ng/mL; 15–15; *p* < 0.001, respectively). Initial uCysB were higher in dogs that died during the one-month follow-up period (n = 10) (731 ng/mL; 517–940), compared to survivors (n = 8) (25 ng/mL; 15–417 (*p* = 0.009). Based on these results, uClust and especially uCysB are promising biomarkers of AKI. Further, they might reflect the severity of tubular injury, which is known to be central to the pathology of AKI.

## 1. Introduction

Acute kidney injury (AKI) varies from mild injury to the nephrons to a severe loss of kidney function. Also, the etiology varies, with common causes in dogs being ischemic/inflammatory causes (such as pancreatitis, peritonitis, sepsis, heat stroke, and acute glomerulonephritis), infectious diseases (such as leptospirosis and pyelonephritis) and exposure to nephrotoxins (such as ethylene glycol, grapes and raisins and over-doses of non-steroidal anti-inflammatory drugs). However, in many cases, the etiology remains unknown [1,2,3,4]. The International Renal Interest Society (IRIS) developed a grading system for AKI in dogs. The grading system is based on an elevation in serum creatinine (sCr) and a decrease in urine output (UO) [5]. However, both biomarkers are suboptimal in detecting AKI in dogs. Serum creatinine increases with a delay from an actual kidney injury and does neither discriminate between ongoing acute or chronic disease nor between primary glomerular or tubular injury [6,7]. Further, sCr is influenced by hydration status and muscle mass [7,8]. Additionally, a precise measurement of UO can be challenging since it requires the placement of indwelling urinary catheters, which are only available for hospitalized patients with a risk of ascending infections [9]. Further, UO can be normal in patients even with severe AKI, making it an insensitive diagnostic tool [1]. Symmetric dimethylarginine (SDMA), another surrogate biomarker of glomerular filtration rate (GFR), shares the limitations of sCr when used for the detection of AKI, except that is not affected by muscle mass [10]. Due to the inherent limitations in the currently available biomarkers, complementary novel diagnostic and prognostic biomarkers are needed in clinical practice.

Urinary clusterin (uClust) and cystatin B (uCysB) elevate in response to tubular damage and could therefore serve as early AKI markers [11]. Clusterin is a protein expressed in numerous tissues, including kidney tissue. However, the glycosylation pattern differs between kidney-specific clusterin and the plasma isoform, making it possible to measure kidney-specific uClust [11]. Clusterin is anti-apoptotic and involved in cell protection, lipid recycling, cell aggregation, and cell attachment. Its detection in urine reflects kidney injuries in both proximal and distal tubules. Less is known about uCysB, a protein found in supernatant collected from ruptured but not from intact, stressed canine renal cells during in vitro studies. In active kidney injury, elevation of uCysB is most likely due to apoptosis and necrosis of renal epithelial cells in the proximal tubules [11].

In healthy dogs, urine concentrations of uClust and uCysB are invariably low, and preliminary data suggests that these could be used as biomarkers to detect renal injury [11,12,13,14,15]. In an observational study, uClust levels were elevated in dogs with AKI caused by leishmaniasis [12]. In an experimental study in dogs, both uClust and uCysB were sensitive in detecting gentamicin-induced renal proximal tubular injury before sCr became elevated [11]. Further, uClust levels were elevated in dogs with hemorrhagic shock treated with colloids [16]. In two recent studies, uClust and uCysB increased in dogs bitten by the European adder, whose venom is suspected to have nephrotoxic properties [13,14]. Additionally, in another recent prospective study, uCysB increased already on the day of surgery in dogs undergoing cardiac surgery under cardiopulmonary bypass, while uClust increased by the second day from surgery in dogs with AKI and likely subclinical AKI [15]. Finally, in a newly published study in dogs with early-stage chronic kidney disease (CKD), uCysB was higher in dogs with progressive disease in comparison to dogs with stable disease, indicating ongoing kidney injury in the former group [17].

Since AKI is multifactorial and the exact etiology is often unknown, biomarkers that are sensitive and specific for the detection of tubular injury at an early stage, irrespective of the underlying etiology, are warranted in clinical practice. The novel biomarkers uClust and uCysB could fulfill these criteria. Thus, our first hypothesis is that uClust and uCysB are elevated in a population of pet dogs with confirmed AKI of different etiology and severity. Our second hypothesis is that the initial concentrations of these biomarkers are directly associated with short-term outcomes in dogs with AKI.

## 2. Materials and Methods

### 2.1. Dogs

Dogs with AKI, irrespective of etiology and healthy control dogs, were recruited for this prospective longitudinal observational study. Included pet dogs with AKI were treated at Cummings School of Veterinary Medicine, Tufts University, MA, USA, between 2017 and 2018. The diagnosis of AKI was based on an acute onset (<14 days) of clinical signs (anorexia, vomiting, diarrhea, lethargy), sCr > 140 µmol/L (1.6 mg/dL) persisting more than 24 h after correction of prerenal factors with concurrent urine specific gravity (USG) < 1.025. Additionally, hospitalized dogs with an increase in sCr ≥ 26 µmol/L (0.3 mg/dL) but with sCr still remaining below > 140 µmol/L over 48 h were eligible for the study. Dogs were classified according to the IRIS AKI grading system into groups I-V based on their initial sCr values after correction of dehydration or hypovolemia [5]. Dogs with known or suspected underlying CKD, a history of long-term PU/PD, previous elevation in sCr, or chronic weight loss were excluded. Other exclusion criteria were a body weight < 4.5 kg and pregnancy. 

Healthy control dogs were recruited amongst staff members’ dogs of the Veterinary Teaching Hospital, University of Helsinki, Finland, in 2019. They were deemed healthy based on unremarkable history and normal physical examination, normal results on complete blood count (CBC), serum biochemistry profile, urinalysis, and urine protein-creatinine ratio (UPC).

### 2.2. Animal Ethics

The study protocol was approved by the Institutional Animal Care and Use Committee (IACUC, G2017-35) at Tufts University &Tufts Medical Center and the Human Nutrition Research Center on Aging, and by the Animal Experiment Board at the Regional State Administrative Agency of Southern Finland (ESAVI/3381/04.10.07/2017). All dog owners signed an informed consent form.

### 2.3. Sample Collection and Follow-Up

Blood and urine samples were collected from AKI dogs within 24 h after arrival at the hospital and thereafter daily for the first four days of hospitalization, then every second day until discharge or death. Urine from dogs with AKI was collected by catheterization, cystocentesis, or voiding. Blood and urine samples from the healthy control dogs were collected at study inclusion using free voiding as a sampling method for urine. Blood and urine samples were collected again in both groups at follow-up visits around 4 weeks and 3 months after hospital discharge in dogs with AKI or after inclusion day in control dogs.

### 2.4. Laboratory Analysis

Laboratory analyses included CBC, analyses of serum biochemistry (glucose, creatinine, urea, phosphorus, total calcium, sodium, potassium, chloride, total protein, albumin, globulin, alanine aminotransaminase, aspartate aminotransferase, alkaline phosphatase, γ-glutamyl transferase, total bilirubin, cholesterol, creatine kinase), SDMA, *Borrelia burgdorferi* (Bb) antibodies, uClust, uCysB and urinalysis. Serum biochemistry was performed at IDEXX Laboratories, North Grafton, MA, USA (dogs with AKI) and IDEXX Laboratories, Ludwigsburg, Germany (healthy control dogs) using similar clinical chemistry analyzers (AU680 Chemistry Analyzer, Beckman Coulter, Brea, CA, USA). Complete blood count was performed with a Sysmex XT-2000iV instrument at IDEXX Laboratories, North Grafton, MA, USA (dogs with AKI) and IDEXX ProCyte Dx at the Veterinary Teaching Hospital, University of Helsinki (healthy control dogs). SDMA was determined using a validated, commercially available, high-throughput immunoassay (IDEXX SDMA Test, IDEXX Laboratories, Inc.).

Urinalysis included dipstick analyses, UPC, measurement of USG by refractometry, and sediment cytology by microscopy at IDEXX Laboratories, North Grafton, MA, USA (dogs with AKI). In healthy control dogs, urine sediment was analyzed using a SediVue Dx Analyzer (IDEXX Laboratories, Inc.) and visual inspection of recorded images. Urinary creatinine was determined by a colorimetric method, Jaffe’s reaction, using picrate at alkaline pH (Beckman Coulter, Inc., Brea, CA, USA). Urinary protein was measured using a pyrogallol red molybdate method (Beckman Coulter, Inc., Brea, CA, USA). Additionally a urine culture was conducted on dogs that had visible bacteria in their urine sediment.

*Borrelia burgdorferi* (Bb) antibodies were determined by ELISA tests (SNAP 4Dx Plus Test or SNAP 4Dx Test, IDEXX Laboratories, Inc., ME, USA) at inclusion and on each follow-up visit in both dogs with AKI and healthy control dogs. When leptospirosis was suspected, the diagnosis was based on high antibody titers (≥1:1600) in unvaccinated dogs using the microscopic agglutination test (MAT) and/or detection of pathogenic leptospires in the urine or blood with PCR. In vaccinated dogs, leptospirosis was diagnosed with a positive urine or blood PCR [18].

Urine for uClust and uCysB analysis was frozen without delay and stored at −80 °C until analyzed. Concentrations of uClust and uCysBB were measured using sandwich format immunoassays as previously described for dogs [11]. The detection limit for uClust was 70 ng/mL and for uCysB 15 ng/mL.

### 2.5. Data Analysis and Statistical Methods

Descriptive statistics were calculated for initial clinical and clinicopathological variables (dogs with AKI and healthy control dogs). The data’s normality was assessed with the Shapiro–Wilk test, and normality plots and non-parametric variables were reported as median and interquartile range (IQR)/range. The two biomarkers (uClust and uCysB) were reported primarily as absolute values (uClust, uCysB) and secondarily as values normalized to the urine concentration of creatinine (uClust/uCr, uCysB/uCr). Values below the detection limits were imputed using the detection limits (70 ng/mL for uClust and 15 ng/mL for uCysB).

Differences between biomarker levels in dogs with AKI and healthy control dogs at inclusion were evaluated with a non-parametric Mann–Whitney *U*-test. For comparison of biomarker levels and severity of AKI based on IRIS grades, we further divided the patients according to their AKI severity at study inclusion into two groups: IRIS AKI II-III and IRIS AKI IV-V. For comparing three groups (two patient groups and one control group), Kruskall–Wallis test was used. Adjusting for multiplicity was conducted using the False Discovery Rate (FDR) correction of *p* values as proposed by Benjamini and Hochberg [19].

Initial biomarker levels of dogs with AKI that died within one month from hospital discharge (non-survivors) were compared to those of dogs surviving one month from discharge (survivors).

*p*-values of < 0.05 were considered as statistically significant. Statistical analyses were performed with the commercial statistical software (GraphPad Prism 10.1.2., Dotmatics Inc., Boston, MA, USA, 2023; SAS system for Windows, version 9.4, SAS Institute Inc., Cary, NC, USA, 2013 and R for Windows, version 3.6.2, R Foundation for Statistical Computing, Vienna, Austria, 2019).

## 3. Results

### 3.1. Study Population

A total of 28 dogs participated in this study: 18 dogs with AKI (male 10/18, female 8/18) and 10 healthy control dogs (male 3/10, female 7/10). Demographic information for both groups is presented in Table 1. The dogs were of the following breeds in the AKI group: eight mixed breed dogs, two Labrador retrievers and one of each of the following: Black and Tan Coonhound, Cavalier King Charles spaniel, Dalmatian, Jack Russel terrier, Old English Bulldog, Rottweiler, Shetland Sheepdog and Staffordshire Bull terrier. The healthy control group consisted of one of each of the following breeds: Belgian Malinois, Cairn Terrier, Finnish Lapphund, German Shepherd, Hovawart, Labrador Retriever, Miniature Pinscher, Standard Schnauzer, Smooth Collie, and Wirehaired Dachshund.

### 3.2. Etiology of AKI

Lyme nephritis was suspected in 7/18 dogs with AKI based on a positive antibody snap test and severe proteinuria (UPC > 4). However, urine bacterial culture was positive in four of these seven dogs, such that pyelonephritis as the primary cause of AKI could not be ruled out. Leptospirosis was diagnosed in 3/18 AKI dogs (positive PCR in 2/3, MAT 1:6400 in an unvaccinated dog, 1/3). Other singular causes of AKI were heatstroke, grape intoxication, sepsis, multi-organ failure, urethral obstruction in combination with possible pyelonephritis, and ibuprofen intoxication. In 2/18 dogs, the etiology of the AKI remained unknown.

### 3.3. Clinicopathological Results

Initial hematocrit, albumin, phosphorus, creatinine, SDMA, UPC, and USG are presented in Table 1. Based on initial sCr, 18 AKI dogs represented the following IRIS AKI grades: grade II n = 1, grade III n = 5, IV n = 11, and grade V n = 1. There were no IRIS AKI grade I dogs [5].

The first blood and urine samples were collected within 24 h after arrival at the hospital in all AKI dogs, except for one severely oliguric dog from which the first urine sample was obtained on day two. In one dog with AKI, initial uCr concentration was not measured/recorded, and this dog’s results are therefore not included in the uCr normalized biomarker results. Twelve (67%) of the dogs with AKI had antibodies against Bb based on a SNAP 4DX test, while none of the healthy control dogs tested positive. Out of the Bb seropositive AKI dogs, 5/12 were not suspected of having Lyme nephritis based on other more likely causes of their AKI (leptospirosis [n = 2], heat stroke [n = 1], sepsis-related AKI [n = 1], and unknown etiology of AKI but absent proteinuria [n = 1]).

### 3.4. Urinary Biomarkers

Dogs with AKI had significantly higher initial concentrations of uClust (ng/mL) and uCysB in comparison to the control group (Table 1, Figure 1). Biomarkers were significantly higher in AKI dogs also with uCr normalization, reported as uClust/uCr and uCysB/uCr ratios (Appendix A). 

Urinary clusterin (ng/mL) was significantly higher in dogs with IRIS AKI II-III (median 4995; range 70–12,173) and IV-V (median 3593, range 378–12,500) than in healthy control dogs (median 70; range 70–587; *p* = 0.03 and *p* < 0.001), respectively. Similarly, uCysB (ng/mL) was significantly higher in dogs with IRIS AKI II-III (median 274; range 15–698) and IV-V (median 620; range 15–1339) than in healthy control dogs (median 15; range 15–24; *p* = 0.007 and *p* < 0.001, respectively). However, comparing IRIS AKI II-III with IRIS AKI IV-V did not show a significant difference in either uClust (*p* = 0.15) or uCysB (*p* = 0.67) (Figure 2). The results were similar for biomarker concentrations normalized with uCr (Appendix A).

### 3.5. Disease Severity and Outcome 

Overall mortality in dogs with AKI was 61% (11/18). Seven (39%) of the dogs with AKI died before hospital discharge and four after hospital discharge (median 7.5 days from discharge, range 4–64 days). All seven dogs with Lyme nephritis died during the study period. Of the eleven deceased dogs, 3/11 died naturally, and 8/11 were euthanized due to ethical reasons and being unresponsive to treatment. 11/18 AKI dogs survived hospital discharge, and their median duration of hospitalization from study inclusion was 4 days (range 2–13 days). 

Four dogs were treated with hemodialysis due to the severity of their clinical signs, uremia, decreased/ceased urine production, or fluid overload. While three of these dogs were alive at the end of the study (two with leptospirosis and one with grape intoxication), one diagnosed with multi-organ failure (MOF) did not survive to hospital discharge.

### 3.6. Follow-Up

Eleven dogs entered the follow-up phase of the study after hospital discharge, but three dogs died before the first control visit. Thus, eight dogs were known to be alive one month after hospital discharge, and six of these were presented to the first follow-up visit at a median of 33 days (range 28–44 days) after discharge. Additionally, one dog died between the first and second follow-up visits, and three dogs were lost to follow-up. Four dogs were thus presented to the second follow-up, which took place at a median of 91.5 days (range 83–105) from discharge. One of these dogs did not attend the first control visit.

Altogether, four dogs with AKI died after hospital discharge (three before and one after follow-up visit 1), and all of these dogs died due to AKI-related reasons.

All ten healthy control dogs attended the planned three visits and did not show signs of disease during this study period. The first visit for the healthy control dogs was 28 days (range 27–34 days), and the second visit was 89 days (range 84–102) from study inclusion.

### 3.7. Biomarkers in Survivors vs. Non-Survivors

A comparison of biomarkers in survivors (n = 8) and non-survivors (n = 10) was performed based on one-month survival since 3/8 one-month survivors were lost to follow-up by the three-month follow-up visit. Initial uCysB concentrations were significantly higher in the non-survivors (n = 10) in comparison to the one-month survivors (n = 8), with uCysB median being 731 ng/mL [IQR 517–940] vs. 25 ng/mL [IQR 15–417] (*p* = 0.009). The difference was non-significant for uClust with a median of 8709 [IQR 2618–11,999] for non-survivors vs. 2223 ng/mL [415–7232] for survivors (*p* = 0.19) (Figure 3a,b). The differences were non-significant for uCr normalized uClust/uCr (*p* = 0.36) and uCysB/uCr (*p* = 0.08). Initial sCr was not significantly higher in non-survivors than in survivors (Figure 3c).

In the AKI dogs examined at follow-up visits (n = 7), both uClust and uCysB levels showed a decreasing trend with time elapsed from anticipated kidney injury, while in the healthy control dogs, biomarker levels remained low throughout the study. Graphs with individual data from all dogs at all sampling points are shown in Appendix A.

## 4. Discussion

The main findings of our study supported our hypothesis since uClust and uCysB were significantly elevated in dogs with confirmed AKI when compared to healthy control dogs. The increase in these biomarkers is likely due to direct renal tubular damage either in the proximal or distal parts of the renal tubules [11]. Our results are in line with prior experimental and clinical studies showing that dogs with AKI or presumed subclinical AKI have higher concentrations of uClust and uCysB compared to healthy dogs [11,12,13,14,15,16]. However, this study with naturally occurring disease comprises patients with AKI of different etiology and severity, thus differing from the earlier contributions. 

Our results suggest uCysB measurement has prognostic value in dogs with AKI, as initial uCysB concentrations were significantly higher in non-survivors compared to those that survived the one-month follow-up. It can be speculated that dogs with higher uCysB levels had more severe tubular damage with worse outcomes. This is a potentially valuable finding since traditional kidney parameters do not necessarily give information on the possible prognosis early in the course of AKI [4]. Similarly, no difference in uClust or sCr between survivors and non-survivors was observed in our study. These findings, however, require further confirmation in larger cohorts of AKI dogs since our study population was relatively small.

This study diagnosed AKI based on elevations in sCr as recommended by IRIS guidelines [5]. Elevations in serum creatinine inversely reflect a drop in GFR; thus, functional disturbances of the kidneys were known to be present in all sick dogs in this cohort. The pathophysiology of AKI is complex and etiology-dependent [20]. Severe tubular damage is, however, needed before GFR decreases markedly [21]. This is due to the compensatory capacity of less and non-affected nephrons aiding to retain GFR [22]. Interestingly, in this study, dogs with AKI of lower IRIS grade did not have significantly lower uClust nor uCysB levels compared to the dogs with higher IRIS AKI grades. One explanation for this observation could be that the IRIS grading system evaluates kidney function, whereas uClust and uCysB reflect structural damage, and that functional and structural changes do not always fully correlate. The groups were, however, small, and thus, this observation needs to be further studied in larger populations for confirmation.

In addition to the differences in uClust, uCysB, and sCr between dogs with AKI and healthy control dogs, also hematocrit, serum albumin, phosphorus, SDMA, UPC, and USG differed between dogs with AKI and healthy dogs. These results reflect typical changes in laboratory parameters during AKI, strengthening the diagnosis [1,4,10]. 

AKI is a life-threatening condition with a substantial risk for mortality [3]. Sixty-one percent of dogs with AKI died either during initial hospitalization or during the three-month study period, which is a higher mortality rate than the reported overall mortality rate of 45% in dogs with AKI in a recent meta-analysis [3]. Most dogs in our study had severe AKI based on IRIS AKI grading, and several patients had marked proteinuria. Previous studies in dogs have shown that high sCr in combination with marked proteinuria carries a poor prognosis [23]. This could explain the low survival rates in our dogs with AKI. However, the number of dogs in this study was small, and the high mortality rate could also have been due to the coincidental inclusion of more severe AKI cases in this study. 

Clinical Lyme nephritis has traditionally been considered a chronic disease with mainly glomerular involvement [24,25,26]. However, dogs with possible Lyme nephritis in our cohort presented with acute signs (<2 weeks reported by the owners and without previous signs of CKD) and thus fit our inclusion criteria. Interestingly, these dogs showed significant increases in the tubular markers uClust and uCysB. This can be explained by the findings of an earlier histopathological study where tubular necrosis was found in addition to glomerular lesions in dogs with a presumptive diagnosis of Lyme nephritis [27]. More specifically, dogs in that study were found to have cortical tubular dilation with multifocal tubular epithelial cell necrosis and regeneration without significant tubular atrophy and interstitial fibrosis typically seen in other glomerulonephropathies [27]. None of the dogs in our study with a presumptive diagnosis of Lyme nephritis (and elevation in uClust and uCysB) survived throughout the three-month study period. 

Even though our aim was to study dogs with AKI, it is possible that some of the dogs (such as the Lyme nephritis dogs) actually had “acute on chronic” kidney disease. This is not necessarily a disadvantage of the study since these patients likely have had an active, ongoing tubular injury on top of their possible CKD, thus being a relevant subpopulation. It has been proposed that CKD might actually be a continuum of repeated AKIs [17,28]. 

The majority of all dogs with AKI in this study, but none of the healthy control dogs, were seropositive for Bb. The explanation for this is that dogs with AKI were recruited in Massachusetts, USA, a highly endemic area of Bb, with a reported seroprevalence of 15.3% [29]. In addition, some of these dogs, as mentioned above, had possible Lyme nephritis. The healthy control dogs were recruited in Southern Finland, where the reported Bb seroprevalence in dogs is only 2.9% [30]. This geographical difference prevents the comparison of uClust and uCysB concentrations in Bb antibody-positive dogs with AKI and asymptomatic Bb antibody-positive dogs in this study. Future studies could investigate whether uClust and uCysB reflect tubular damage in dogs with Lyme nephritis and whether this could be used to predict the outcome of Bb antibody-positive dogs before developing clinical signs of renal disease.

Because of the high mortality rate and the number of surviving dogs lost to follow-up, advanced statistical analyses were not performed regarding uClust and uCysB levels over time, and only descriptive data was reported. Decreasing concentrations of both uClust and uCysB from study inclusion to the last control visit were observed in surviving dogs with AKI, while biomarker levels remained consistently low in the healthy control dogs, as expected. This finding supports the hypothesis that uClust and uCysB reflect active, ongoing tubular damage, which was also seen in previous experimental and observational studies [11,12,13,14,15,16,17]. 

We reported uClust and uCysB primarily as not normalized to uCr (normalized results did not differ from the non-normalized main results). The reason for this approach is that normalization with uCr is problematic when used in AKI, where the excretion of uCr can vary substantially. When there is a reduction in excreted creatinine as a result of GFR decline, uCr normalized urinary biomarker levels will be overestimated. On the other hand, when GFR increases again in the recovery phase of AKI, and more creatinine is excreted in the urine, the reduction in urinary biomarker levels corrected with uCr is exaggerated [31].

A limitation of this study is that when measuring biomarkers in serum samples of critically ill patients, we cannot be certain that uClust and uCysB originated solely from the kidneys. Even though uCysB and uClust are known to be excreted from renal tubules during injury, their metabolism is not fully understood, and therefore it is possible that excretion from other organs exists, adding up to the urinary concentrations noted. Including another control group of dogs with multisystemic disorders without AKI would have helped to assess these biomarkers’ specificity in addition to sensitivity. 

Another limitation of the study is that the diagnosis of leptospirosis was based on a single high MAT in one unvaccinated dog, which is not fully reliable [32]. However, as this study aimed not to compare biomarkers among different etiologies of AKI, this uncertainty in etiology does not affect the interpretation of the results.

Additionally, a limitation of this study is the recruitment of dogs with AKI and healthy control dogs in different geographical locations, such as the USA and Finland. Even so, uClust and uCysB levels in the healthy control dogs were consistent with biomarker levels of other control populations [11,12,13,14,15]. 

## 5. Conclusions

We demonstrated that two novel tubular biomarkers, uClust and uCysB, were significantly increased in dogs with AKI of different etiology and severity in comparison to healthy control dogs and show thus promise as biomarkers of AKI. 

In addition, initial levels of uCysB, but not sCr, were lower in short-term survivors compared to non-survivors, indicating that this novel biomarker might have a prognostic value irrespective of sCr.

## Figures and Tables

**Figure 1 vetsci-11-00200-f001:**
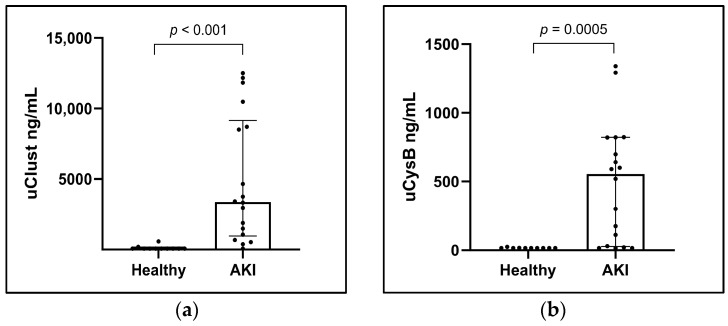
Column scatter plots of (**a**) urinary clusterin (uClust) and (**b**) urinary cystatin B (uCysB) in healthy control dogs (Healthy) (n = 10) and with acute kidney injury (AKI) (n = 18) at inclusion visit. Inclusion visit: <24 h from arrival at hospital (dogs with AKI) or study inclusion (control dogs).

**Figure 2 vetsci-11-00200-f002:**
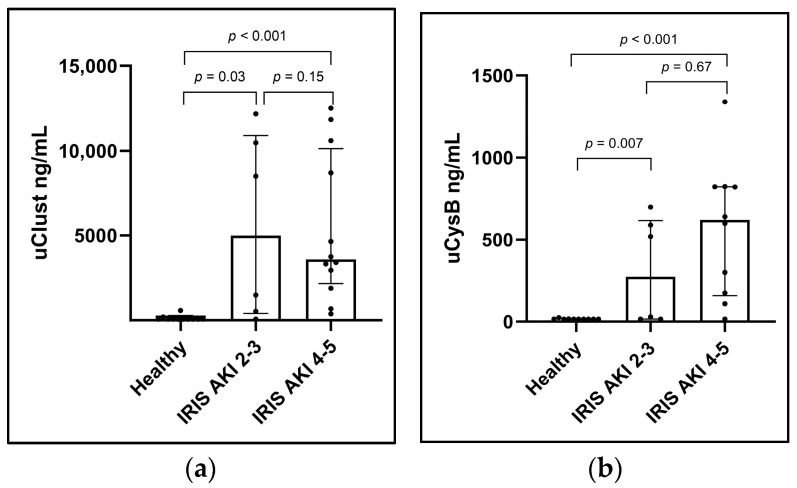
Column scatter plots showing (**a**) urinary clusterin (uClust) and (**b**) urinary cystatin B (uCysB) by initial acute kidney injury (AKI) grade. AKI grades are based on the International Renal Interest Society (IRIS) AKI grading system on inclusion visits. Number of dogs in each group: healthy control dogs (Healthy) n = 10; IRIS AKI 2–3, n = 6; IRIS AKI 4–5, n = 12. Inclusion visit: <24 h from arrival at hospital (dogs with AKI) or study inclusion (healthy control dogs).

**Figure 3 vetsci-11-00200-f003:**
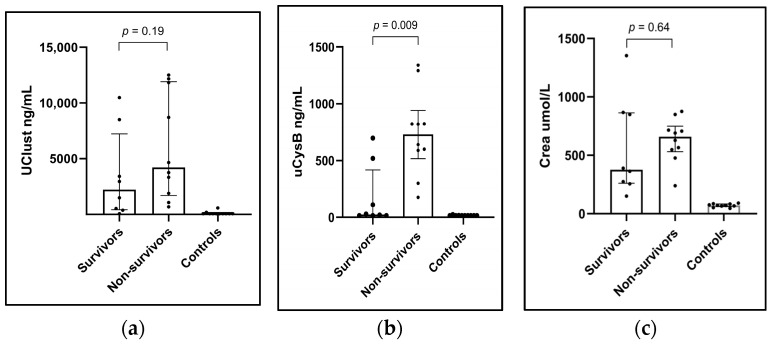
Column scatter plots showing showing (**a**) urinary clusterin (uClust), (**b**) urinary cystatin B (uCysB), and (**c**) serum creatinine (Crea) concentrations at the inclusion visit in dogs with acute kidney injury (AKI) surviving one-month (Survivors) (n = 8), non-survivors (n = 10) and healthy control dogs (n = 10). Inclusion visit: <24 h from arrival at the hospital (dogs with AKI) or study inclusion (healthy control dogs).

**Table 1 vetsci-11-00200-t001:** Demographic and clinicopathological data for dogs with acute kidney injury (AKI) and healthy control dogs at inclusion to the study.

Parameter (Reference Range)	Healthy (n = 10) Median (Range)	AKI (n = 18) Median (Range)	*p*
Age (years)	6.0 (2.5–10.4)	8.1 (0.4–13.3)	0.31
Body weight (kg)	22.5 (4.9–30.0)	27.5 (6.5–40.4)	0.10
Hematocrit (%) (38.3–56.5)	48.5 (42.5–57.6)	34.3 (25.1–51.0)	0.001
Albumin (g/L) (27–39)	32 (29–37)	21 (13–36)	<0.001
Phosphorus (mmol/L) (0.8–2.0)	1.3 (1.1–1.9)	3.8 (0.9–7.2)	<0.001
Creatinine (µmol/L) (44–133)	71 (44–88)	557 (150–1353)	<0.001
SDMA (ug/dL) (0–14) (0–16 in puppies)	10.1 (7.4–13.3)	60.5 (9.0–104.0)	<0.001
UPC (<0.5)	0.3 (0.1–0.4)	3.8 (0.5–23.9)	0.03
USG	1.039 (1.007–1.049)	1.012 (1.008–1.024)	0.013
uClusterin (ng/mL)	70 (70–587)	3593 (70–12,500)	<0.001
uClusterin/uCreatinine (×10^−6^)	54 (21–326)	5750 (737–22,665) *	<0.001
uCystatin B (ng/mL)	15 (15–24)	554 (15–1339)	<0.001
uCystatin B/uCreatinine (×10^−6^)	10 (5–40)	917 (32–5463) *	<0.001

SDMA, symmetric dimethyl arginine; UPC, urine protein-to-creatinine ratio; USG, urine specific gravity; uClusterin, urinary Clusterin; uCystatin B, urinary Cystatin B; uCreatinine, urinary Creatinine. * n = 17/18.

## Data Availability

The datasets used and analyzed during the current study are available from the corresponding author upon a reasonable request.

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
