# Peer review of "A Clinical Study on Urinary Clusterin and Cystatin B in Dogs with Spontaneous Acute Kidney Injury"

_vetsci, 2024, doi:10.3390/vetsci11050200_

Round 1

Reviewer 1 Report

Comments and Suggestions for Authors

I would like the authors to describe in their next study more accurate values of the investigated indicators of acute renal failure in different diagnoses and if possible to include more patients

Author Response

Thank you for the comment, we fully agree that larger and more accurate studies are needed in the future. 

Reviewer 2 Report

Comments and Suggestions for Authors

I appreciate the opportunity to review this manuscript. The manuscript presents a Study on Urinary Clusterin and Cystatin B in Dogs as biomarkers of Spontaneous Acute Kidney Injury. This is an interesting clinical study with relevant conclusions with uClust and uCysB being seen as promising markers acute kidney injury for diagnosing and predicting the outcome of AKI in dogs. I have only minor remarks

Line 42: Although AKI grading system has been accepted in 2023, it has been developing for some time. Please correct.
Line 122: Please discriminate serum biochemistry analyses.

Line 140: Discriminating between SNAP 4Dx Plus and SNAP 4Dx Test, may be misleading for readers

Line 123 and 143: For a better understanding on the part of the reader, I think it will be better to introduce Lyme disease, leptospirosis, as well as other etiologies of AKI, in the introduction section. Line 143/4: Explain the differences on the leptospirosis diagnosis protocol in unvaccinated dogs. Line 178: table 1 doesn’t refer to demographic  information but clinicopathological data for dogs with AKI and control dogs   Table 1: I think the first line with the columns identification is missing  

Author Response

Line 42: Although AKI grading system has been accepted in 2023, it has been developing for some time. Please correct.

Author: Thank you for this comment, we are aware of the IRIS grading system being developed already earlier (in 2012-2013), and the grading that we referred to in our article is from 2016. We have now removed the word “recently” and added the year “2016” to the reference. This can be found on Line 48.

Line 122: Please discriminate serum biochemistry analyses.

Author: We have now added the serum biochemistry analyzes that were performed in all dogs. These can be found on lines 129-132.

Line 140: Discriminating between SNAP 4Dx Plus and SNAP 4Dx Test, may be misleading for readers

Author: The reason for discriminating the two tests is that they are not exactly similar (the Snap 4DX Plus measures Ehrlichia ewingii and Anaplasma platys antibodies in addition to Anaplasma phagocytophilum, Ehrlichia canis, Ehrlichia chaffeensis, Borrelia burgdorferi and heartworm antibodies while Snap 4DX only measures the latter ones). At the time of the study, the Snap 4DX test was the only one available in Finland while Snap4DX plus was used in the US. The text has however now been changed and we hope this will improve the readability. This can be found on line 151.

Line 123 and 143: For a better understanding on the part of the reader, I think it will be better to introduce Lyme disease, leptospirosis, as well as other etiologies of AKI, in the introduction section.

Author: Thank you, we agree on the comment and have now added a few sentences about the etiologies of AKI in dogs in the introduction. These can be found on lines 42-47.

Line 143/4: Explain the differences on the leptospirosis diagnosis protocol in unvaccinated dogs.

Author: Thank you, we have added information about how leptospirosis was diagnosed in this study as well as one limitation in the diagnosis of leptospirosis in one of the AKI dogs. This can be found on lines 152-156 and 395- 398.

Line 178: table 1 doesn’t refer to demographic information but clinicopathological data for dogs with AKI and control dogs   Table 1: I think the first line with the columns identification is missing  

Author: Table 1: We think Table one´s layout changed during editing and also the first line had disappeared. We have now corrected it back to the original version and hope it is clearer.

Reviewer 3 Report

Comments and Suggestions for Authors

Line 56: Are there 'baseline/normal' amounts for these substances? Do these amounts change with age? 

Line 69: Is there any information on how fast these markers elevate? 

Table 1: Please add labels to the tops of the columns.

Line 241: With this small sample you may not be able to make conclusions, but were the Lyme dogs values higher/lower/similar to other causes.  Any difference between infectious and toxic?

Nice discussion about limitations of study.

Author Response

Line 56: Are there 'baseline/normal' amounts for these substances? Do these amounts change with age? 

Author: The concentration of urinary clusterin and cystatin B  have in several studies been shown to be very low in healthy dogs of different age and breed.  We have added some references to support this statement. These can be found on lines 72-74.

Line 69: Is there any information on how fast these markers elevate?

Author: This is a very important question. It looks like the new biomarkers, especially uCysB elevates very fast from injury (within hours) and we have now added more information from a recently published article about this. This can be found on line 80.

Table 1: Please add labels to the tops of the columns.

Author: Table 1: Labels were missing due to an error in the editing and this has now been corrected.

Line 241: With this small sample you may not be able to make conclusions, but were the Lyme dogs values higher/lower/similar to other causes.  Any difference between infectious and toxic?

Author: This is a very interesting question. However, because of the small sample size and few dogs in each etiology group, we prefered not to perform statistical analyses. We think that investigating larger groups of dogs with AKI of specific etiologies are important topics for future exploration.